# DAG DECORATION:
# CONTINUOUS OPTIMIZATION FOR STRUCTURE LEARNING UNDER HIDDEN CONFOUNDING

## ABSTRACT

We study structure learning for linear Gaussian SEMs in the presence of latent confounding. Existing continuous methods excel when errors are independent, while deconfounding-first pipelines rely on pervasive factor structure or nonlinearity. We propose DECOR, a single likelihood-based and fully differentiable estimator that jointly learns a DAG and a correlated noise model. Our theory gives simple sufficient conditions for global parameter identifiability: if the mixed graph is bow free and the noise covariance has a uniform eigenvalue margin, then the map from $(\mathbf{B}, \mathbf{\Omega})$ to the observational covariance is injective, so both the directed structure and the noise are uniquely determined. The estimator alternates a smooth-acyclic graph update with a convex noise update and can include a light bow complementarity penalty or a post hoc reconciliation step. On synthetic benchmarks that vary confounding density, graph density, latent rank, and dimension with $n < p$, DECOR matches or outperforms strong baselines and is especially robust when confounding is non-pervasive, while remaining competitive under pervasiveness.

## 1 INTRODUCTION

Directed graphical models, especially directed acyclic graphs (DAGs), provide a powerful formalism for representing causal relationships among variables in domains such as biology, economics, and the social sciences (Pearl, 2009; Spirtes et al., 2000b). However, learning the underlying DAG structure from purely observational data remains a fundamental challenge. Even under the linear Gaussian structural equation model (SEM), the observational distribution is generally consistent with an entire Markov equivalence class of DAGs—distinct graphs that encode the same set of conditional independencies (Chickering, 2002b; Andersson et al., 1997). Consequently, without further assumptions or interventional data, the true causal structure is unidentifiable (Squires & Uhler, 2023).

Although linear Gaussian SEMs are generally identifiable only up to a Markov equivalence class, a notable exception occurs when all error terms share the same variance under causal sufficiency the true DAG is identifiable from purely observational data (Peters & Bühlmann, 2014). Outside this equal-variance regime, identifiability typically requires additional asymmetries in the data-generating process, such as non-Gaussian noise (e.g., LiNGAM) or suitable nonlinear additive-noise structure (Shimizu et al., 2006; Hoyer et al., 2008). The challenge is further compounded by latent confounders: unobserved variables can induce spurious associations among observed nodes and destroy identifiability for DAGs, shifting the target to partial ancestral or maximal ancestral graphs and PAGs (Richardson & Spirtes, 2002; Spirtes et al., 2000b; Zhang, 2008). Consequently, learning identifiable causal structure from Gaussian observational data in the presence of latent confounding remains largely open in full generality.

Despite these obstacles, a wave of continuous-optimization methods, initiated by NOTEARS, has advanced DAG discovery from observational data (Zheng et al., 2018). These approaches replace the combinatorial acyclicity constraint with a smooth surrogate (e.g., $h(B) = 0$ based on a matrix exponential), enabling gradient-based minimization of a likelihood- or score-based objective with sparsity regularization. Follow-ups extend the template to various directions (Zheng et al., 2020; Yu

et al., 2019; Lachapelle et al., 2019; Brouillard et al., 2020; Bello et al., 2022). Likelihood-centric variants, such as GOLEM, make the connection explicit by optimizing the Gaussian (equal- or non-equal-variance) log-likelihood under the smooth acyclicity constraint (Ng et al., 2020). Across this family, a common assumption is causal sufficiency (no unmeasured confounding) with mutually independent noise terms; in practice, violations of this assumption, e.g., latent confounders—can bias edge orientation and degrade recovery (Spirtes et al., 2000b).

Complementary progress on handling hidden confounding has emerged along two fronts. First, methods that exploit distributional asymmetries in non-Gaussian models build on the LiNGAM paradigm (Shimizu et al., 2006). A particularly useful structural assumption in this regime is *bow-freeness*, which forbids any unordered pair of observed variables from carrying both a directed edge and a bidirected error link. Bow-free constraints yield identifiability results for mixed graphs in the non-Gaussian setting, and recent work leverages this to orient edges and detect latent siblings without prior knowledge of the number or placement of confounders (Wang & Drton, 2023). Related results establish parameter identifiability—of edge coefficients and noise covariances—under linear Gaussian SEMs on acyclic mixed graphs, including generalized bow-free structures; in particular, Drton et al. (2011).

Second, deconfounding-first strategies estimate latent influences before DAG discovery proceeds. In this pipeline, one first recovers low-dimensional latent structure from observational data—using factor or spectral methods, principal components, or low-rank plus sparse decompositions—then removes the estimated confounding signal prior to causal graph learning (Frot et al., 2019; Shah et al., 2020; Agrawal et al., 2023; Squires et al., 2022; Chandrasekaran et al., 2010). These approaches typically rely on a *pervasive confounding* assumption, namely that a small number of latent factors load on many observed variables with non-negligible strength, which makes the confounding component identifiable by PCA-type estimators.

Despite this progress, a gap remains. To our knowledge, no continuous-optimization approach both removes latent confounding and learns the DAG in linear Gaussian SEMs when confounding is non-pervasive, that is, when latent factors do not load broadly across many variables. Existing smooth-acyclicity methods typically assume causal sufficiency, and deconfounding-first pipelines rely on pervasive factor structure for identifiability.

1. **Parameter identifiability for bow-free mixed graphs.** We establish sufficient conditions for global parameter identifiability of linear Gaussian SEMs with correlated errors. If the mixed graph is bow-free and the error covariance satisfies a uniform eigenvalue margin, then the co-variance parametrization $(\mathbf{B}, \mathbf{\Omega}) \mapsto \mathbf{\Sigma}$ is injective: within the bow-free equivalence class, both the directed coefficients and noise covariance are uniquely determined by the observational co-variance. Our conditions cover both pervasive and non-pervasive confounding patterns without requiring explicit low-rank factorization of the error covariance.

2. **Optimization guarantees from identifiability.** We show that the same structural conditions ensuring identifiability—bow-freeness and the eigenvalue margin—also imply favorable opti-mization geometry. Specifically, these conditions guarantee that the population loss satisfies blockwise restricted strong convexity with bounded cross-block gradients, enabling local con-vergence at the minimax rate $O_p(\sqrt{(s \log p)/n})$. A post-hoc bow projection then recovers exact support under standard signal-strength conditions.

3. **A modular continuous optimization framework.** We develop DECOR, a likelihood-based method that jointly estimates the DAG and a structured error covariance via proximal alter-nating minimization. The procedure alternates between updating the directed graph under a smooth acyclicity constraint with sparsity penalties, and updating the noise covariance within a parametrization respecting the eigenvalue margin. This blockwise design is modular: one can pair any gradient-based DAG optimizer with any compatible covariance estimator.

4. **Integrated deconfounding and discovery.** DECOR replaces the usual two-stage pipeline—first remove confounding, then learn a DAG—with a single estimator that removes latent correlations while orienting edges. Under our identifiability conditions, this yields consistent structure recov-ery and improves robustness when confounding is sparse or localized rather than pervasive.

5. **Empirical validation.** Across synthetic and semi-synthetic benchmarks, DECOR matches or outperforms baselines from smooth-acyclicity methods, constraint- and score-based approaches for latent variables, and deconfounding-first pipelines, over a range of confounding regimes.

## 1.1 PROBLEM FORMULATION

We consider $p$ observed variables indexed by $V = \{1, \ldots, p\}$ generated by a linear Gaussian SEM with possibly correlated errors:

$$\mathbf{x} = \mathbf{B}^\top \mathbf{x} + \mathbf{e}, \qquad \mathbf{e} \sim \mathcal{N}(\mathbf{0}, \mathbf{\Omega}), \qquad \text{acyclicity: } h(\mathbf{B}) = 0, \tag{1}$$

where $\mathbf{B} \in \mathbb{R}^{p \times p}$ is the weighted adjacency matrix of a directed acyclic graph (DAG), $\mathbf{\Omega} \in \mathbb{R}^{p \times p}$ is a positive definite noise covariance, and $h(\mathbf{B}) = 0$ is the smooth NOTEARS constraint that enforces acyclicity (Zheng et al., 2018). The implied covariance is $\mathbf{\Sigma}(\mathbf{B}, \mathbf{\Omega}) = (\mathbf{I} - \mathbf{B}) \, \mathbf{\Omega} \, (\mathbf{I} - \mathbf{B})^\top$.

The directed graph induced by $\mathbf{B}$ has edge set $E^{\rightarrow} \triangleq \mathrm{supp}(\mathbf{B}) = \{i \rightarrow j : B_{ij} \neq 0\}$. The off-diagonal entries of $\mathbf{\Omega}$ encode *bidirected* edges $E^{\leftrightarrow} \triangleq \mathrm{supp}_{\mathrm{off}}(\mathbf{\Omega}) = \{i \leftrightarrow j : \Omega_{ij} \neq 0, i \neq j\}$, representing latent confounding or correlated noise between $i$ and $j$, Figure 1c. Together, the directed and bidirected edges form an *acyclic mixed graph* $G = (V, E^{\rightarrow}, E^{\leftrightarrow})$ (Drton et al., 2011). Throughout, we distinguish between

- *parameters* $(\mathbf{B}, \mathbf{\Omega})$, which fully specify the linear-Gaussian SEM;
- the induced observational distribution $\mathbf{x} \sim \mathcal{N}(\mathbf{0}, \mathbf{\Sigma}(\mathbf{B}, \mathbf{\Omega}))$;
- the associated mixed graph $G(\mathbf{B}, \mathbf{\Omega}) = (V, E^{\rightarrow}(\mathbf{B}), E^{\leftrightarrow}(\mathbf{\Omega}))$.

In general, the map $(\mathbf{B}, \mathbf{\Omega}) \mapsto \mathbf{\Sigma}(\mathbf{B}, \mathbf{\Omega})$ is many-to-one: different mixed graphs (structural differences) and different parameter pairs (even for the same structure) can yield the same observational covariance. Understanding and addressing this non-identifiability is central to our theory.

Given $n$ i.i.d. samples $\mathbf{x}_1, \ldots, \mathbf{x}_n \sim \mathcal{N}(\mathbf{0}, \mathbf{\Sigma})$ arranged as rows of $\mathbf{X} \in \mathbb{R}^{n \times p}$, a negative log-likelihood for $(\mathbf{B}, \mathbf{\Omega})$, up to additive constants, is

$$\mathcal{L}_n(\mathbf{B}, \mathbf{\Omega}) = \frac{1}{n} \left\| \mathbf{\Omega}^{-1/2} (\mathbf{X} - \mathbf{X}\mathbf{B}) \right\|_F^2 + \log \det \mathbf{\Omega} - 2 \log \det(\mathbf{I} - \mathbf{B}). \tag{2}$$

**Connections to existing objectives.** (i) If $\mathbf{\Omega} = \mathbf{I}$ and $h(\mathbf{B}) = 0$ enforces a DAG, then under a topological ordering $\det(\mathbf{I} - \mathbf{B}) = 1$, so $\mathcal{L}_n$ reduces to least squares on the residuals plus a constant. With sparsity regularization on $\mathbf{B}$, this recovers the NOTEARS objective (Zheng et al., 2018). (ii) The GOLEM family optimizes the Gaussian likelihood *without* a separate smooth acyclicity penalty, instead converting the last term to a regularizer $-2\log|\det(\mathbf{I} - \mathbf{B})|$, which vanishes for DAGs (Ng et al., 2020).

Our proposed algorithm, DECOR, minimizes a penalized version of equation 2, adding sparsity-inducing penalties on $\mathbf{B}$ and on either $\mathbf{\Omega}$ or $\mathbf{\Theta} \triangleq \mathbf{\Omega}^{-1}$, together with the smooth acyclicity penalty $h(\mathbf{B})$:

$$\mathcal{J}_n(\mathbf{B}, \mathbf{\Omega}) = \mathcal{L}_n(\mathbf{B}, \mathbf{\Omega}) + \lambda_B \|\mathbf{B}\|_1 + \lambda_\Omega \, \mathrm{pen}_\Omega(\mathbf{\Omega}) + \rho \, h(\mathbf{B}). \tag{3}$$

Our main algorithmic contribution (Section 4) is a scalable alternating minimization scheme over $(\mathbf{B}, \mathbf{\Omega})$ that preserves the favorable structure of equation 2 and jointly learns directed and confounded structure.

**Mixed-graph notation.** For later use, we adopt standard mixed-graph notation. For a node $i$, let $\mathrm{Pa}(i) \triangleq \{j : B_{ji} \neq 0\}$ denote its directed parents and $\mathrm{Sib}(i) \triangleq \{j : \Omega_{ij} \neq 0, j \neq i\}$ its bidirected neighbors (siblings). We write $[i] = \{1, \ldots, i\}$ after a topological ordering of the DAG, and for a matrix $M$, $M_{R,C}$ denotes the submatrix with rows in $R$ and columns in $C$.

## 2 IDENTIFIABILITY, EQUIVALENCE CLASSES, AND BOW-FREE MIXED GRAPHS

We now develop the identifiability theory underlying DECOR. Our goals are twofold: (i) understand when the SEM parameters $(\mathbf{B}, \mathbf{\Omega})$ are uniquely determined

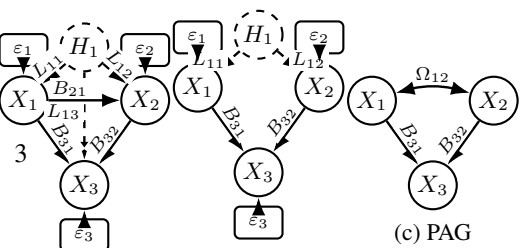

(c) PAG

by the observational covariance *within a fixed mixed graph*; and (ii) characterize the larger equivalence class of mixed graphs that induce the same observational distribution, and how sparsity-based regularization selects canonical representatives. For the first goal, our key contribution is showing that *bow-freeness plus an eigenvalue margin* yields a simple sufficient condition for parameter identifiability, making the general framework of Drton et al. (2011) amenable to continuous optimization. For the second, we connect this to a sparsity-based minimal equivalence class in the spirit of Deng et al. (2024), extending their results from DAGs with diagonal noise to acyclic mixed graphs with general noise covariance.

## 2.1 PARAMETER IDENTIFIABILITY ON A FIXED BOW-FREE MIXED GRAPH

Fix an acyclic mixed graph $G = (V, E^\rightarrow, E^\leftrightarrow)$ specifying the connectivity structure, and let

$$\mathcal{M}(G) := \left\{ (\mathbf{B}, \mathbf{\Omega}) : \mathbf{B} \text{ acyclic}, \ \mathbf{\Omega} \succ 0, \ \mathrm{supp}(\mathbf{B}) = E^\rightarrow, \ \mathrm{supp}_{\mathrm{off}}(\mathbf{\Omega}) = E^\leftrightarrow \right\}$$

be the parameter space indexed by $G$, and define the covariance map

$$\phi_G : \mathcal{M}(G) \to \mathbb{S}^p_{++}, \qquad \phi_G(\mathbf{B}, \mathbf{\Omega}) \ = \ \mathbf{\Sigma}(\mathbf{B}, \mathbf{\Omega}) \ = \ (\mathbf{I} - \mathbf{B})^{-1} \mathbf{\Omega} (\mathbf{I} - \mathbf{B})^{-\top}.$$

Following Drton et al. (2011), we say that the model $\mathcal{M}(G)$ is *globally identifiable* if $\phi_G$ is injective: whenever $(\mathbf{B}, \mathbf{\Omega}), (\mathbf{B}', \mathbf{\Omega}') \in \mathcal{M}(G)$ satisfy $\mathbf{\Sigma}(\mathbf{B}, \mathbf{\Omega}) = \mathbf{\Sigma}(\mathbf{B}', \mathbf{\Omega}')$, we have $(\mathbf{B}, \mathbf{\Omega}) = (\mathbf{B}', \mathbf{\Omega}')$. This is *parameter identifiability conditional on the graph*: it guarantees uniqueness of edge weights and noise covariances within $G$, but does not preclude a *different* mixed graph $G'$ with its own parameters from generating the same covariance.

**Drton–Foygel–Sullivant rank condition.** Drton et al. (2011) show that global identifiability of $\mathcal{M}(G)$ is equivalent to the absence of a certain graphical obstruction[1] and can be expressed as a nodewise rank test. Let $\mathbf{T} = (\mathbf{I} - \mathbf{B})^{-1}$ denote the total effect matrix, which captures both direct and indirect causal effects. Then $\mathcal{M}(G)$ is globally identifiable if and only if, for every node $i$ with $\mathrm{Pa}(i) \neq \emptyset$,

$$\mathrm{rank}\left( \mathbf{\Omega}_{[i] \setminus \mathrm{Sib}(i), [i]} \ \mathbf{T}_{[i], \mathrm{Pa}(i)} \right) = |\mathrm{Pa}(i)| \tag{4}$$

holds for all $(\mathbf{B}, \mathbf{\Omega}) \in \mathcal{M}(G)$, where $[i] = \{1, \ldots, i\}$ under a topological ordering. Intuitively, the left factor removes rows corresponding to nodes that share confounding with $i$, while the right factor carries the causal signatures of the parents; when their product has full column rank, parent effects are distinguishable from confounding.

**Bow-freeness and eigenvalue margin as sufficient conditions.** Directly verifying equation 4 requires checking all parameter values and scanning induced subgraphs which is impractical for optimization. Our first theoretical contribution provides *simple structural and spectral conditions* that guarantee equation 4, making the DFS framework compatible with continuous optimization.

**Definition 2.1** (Bow-freeness). An acyclic mixed graph $G = (V, E^\rightarrow, E^\leftrightarrow)$ is *bow-free* if for every unordered pair $\{i, j\}$, at most one of the following holds: $i \rightarrow j \in E^\rightarrow$, $j \rightarrow i \in E^\rightarrow$, or $i \leftrightarrow j \in E^\leftrightarrow$.

**Definition 2.2** (Eigenvalue margin). A positive definite matrix $\mathbf{\Omega} \succ 0$ satisfies an *eigenvalue margin* with constant $m > 0$ if $\mathbf{\Omega} \succeq m\mathbf{I}$.

---

[1] A *converging arborescence* is a directed subtree in which all edges point toward a single node called the sink. The obstruction occurs when such a structure is "covered" by a connected bidirected component—meaning every non-sink node in the arborescence shares a bidirected edge with another node in the same structure. The simplest instance is a *bow*: a pair $\{i, j\}$ with both $i \rightarrow j$ and $i \leftrightarrow j$, which is precisely what bow-freeness forbids.

Bow-freeness ensures that $\mathrm{Pa}(i) \cap \mathrm{Sib}(i) = \emptyset$ for every node $i$: no variable can be simultaneously a parent and a sibling. This prevents directed and bidirected effects from colliding on the same pair and guarantees that removing sibling rows in equation 4 preserves all parent rows. The eigenvalue margin ensures that $\mathbf{\Omega}$ remains well-conditioned, so the left factor in equation 4 retains full row rank with a quantitative singular value bound.

**Lemma 2.3.** *Let $(\mathbf{B}, \mathbf{\Omega}) \in \mathcal{M}(G)$ with $\mathbf{B}$ acyclic, and let $\mathbf{T} = (\mathbf{I} - \mathbf{B})^{-1}$. Under any topological ordering, $\mathbf{T}$ is unit lower triangular. For each node $i$, the submatrix $\mathbf{T}_{[i],\,\mathrm{Pa}(i)}$ has full column rank with $\sigma_{\min}(\mathbf{T}_{[i],\,\mathrm{Pa}(i)}) \geq 1$.*

**Lemma 2.4.** *If $\mathbf{\Omega} \succeq m\mathbf{I}$ for some $m > 0$, then for every node $i$, the submatrix $\mathbf{\Omega}_{[i]\setminus\mathrm{Sib}(i),\,[i]}$ has full row rank with smallest singular value at least $m$.*

Combining Lemmas 2.3 and 2.4, yields a simple sufficient condition for DFS rank condition of Equation 4 which results in the following global parameter identifiability theorem.

**Theorem 2.5** (Sufficient conditions for global identifiability). *Let $G = (V, E^{\rightarrow}, E^{\leftrightarrow})$ be a bow-free acyclic mixed graph, and let $\mathcal{M}_m(G) := \{(\mathbf{B}, \mathbf{\Omega}) \in \mathcal{M}(G) : \mathbf{\Omega} \succeq m\mathbf{I}\}$ denote the parameter space restricted to noise covariances with eigenvalue margin $m > 0$. Then the rank condition equation 4 holds at every node $i$ with $\mathrm{Pa}(i) \neq \emptyset$ and for all $(\mathbf{B}, \mathbf{\Omega}) \in \mathcal{M}_m(G)$. Consequently, the covariance map $\phi_G$ is injective on $\mathcal{M}_m(G)$, so distinct parameters $(\mathbf{B}, \mathbf{\Omega})$ yield distinct observational distributions $\mathcal{N}(\mathbf{0}, \mathbf{\Sigma}(\mathbf{B}, \mathbf{\Omega}))$.*

We assume the true SEM parameters in Equation 1 satisfy the sufficiency conditions of Theorem 2.5:

**Assumption 2.6** (Bow-freeness). The true mixed graph $G(\mathbf{B}^{\star}, \mathbf{\Omega}^{\star})$ is bow-free.

**Assumption 2.7** (Eigenvalue margin). The true noise covariance satisfies $\mathbf{\Omega}^{\star} \succeq m\mathbf{I}$, $m > 0$.

Under these assumptions, Theorem 2.5 implies that given the structure of $G^{\star}$ the mapping $\mathcal{M}(G^{\star})$ is globally identifiable: the directed coefficients and noise covariance are uniquely determined by the observational covariance. Importantly, both conditions are compatible with continuous optimization as we explain in Section **??**.

## 2.2 EQUIVALENCE CLASSES AND SPARSEST BOW-FREE REPRESENTATIONS

Theorem 2.5 establishes parameter identifiability *within* a fixed bow-free mixed graph $G$: if $G$ is bow-free and the eigenvalue margin holds, then no two distinct parameter pairs in $\mathcal{M}(G)$ produce the same covariance. However, this does not preclude the existence of *other* bow-free mixed graphs $G' \neq G$ with their own parameters $(\mathbf{B}', \mathbf{\Omega}')$ that yield the same observational distribution. Such distributionally equivalent bow-free graphs are known to exist in the Gaussian setting (Nowzohour et al., 2017; van Ommen & Mooij, 2017).

**Bow-free equivalence classes.** Let $\mathbf{\Sigma}^{\star} = \mathbf{\Sigma}(\mathbf{B}^{\star}, \mathbf{\Omega}^{\star})$ denote the true covariance. We define the *bow-free equivalence class*

$$\mathcal{E}^{\mathrm{bow}}(\mathbf{\Sigma}^{\star}) := \{(\mathbf{B}, \mathbf{\Omega}) : \mathbf{B} \text{ acyclic}, \ \mathbf{\Omega} \succ 0, \ (\mathbf{B}, \mathbf{\Omega}) \text{ bow-free}, \ \mathbf{\Sigma}(\mathbf{B}, \mathbf{\Omega}) = \mathbf{\Sigma}^{\star}\}.$$

This class collects all bow-free parameter pairs that are observationally indistinguishable from the truth. When $\mathbf{\Omega}$ is restricted to be diagonal (no latent confounding), each element of $\mathcal{E}^{\mathrm{bow}}(\mathbf{\Sigma}^{\star})$ corresponds to a DAG, and the class reduces to the Markov equivalence class studied in Loh & Bühlmann (2014) and Deng et al. (2024). With correlated errors, $\mathcal{E}^{\mathrm{bow}}(\mathbf{\Sigma}^{\star})$ is richer: directed and bidirected edges can trade off while preserving the covariance, though bow-freeness constrains these trade-offs.

Importantly, under the conditions of Theorem 2.5, $\mathcal{E}^{\mathrm{bow}}(\mathbf{\Sigma}^{\star})$ is finite: there are only finitely many bow-free mixed graphs on $p$ nodes, and for each such graph $G$, global identifiability of $\mathcal{M}(G)$ implies that at most one parameter pair $(\mathbf{B}, \mathbf{\Omega}) \in \mathcal{M}(G)$ can satisfy $\mathbf{\Sigma}(\mathbf{B}, \mathbf{\Omega}) = \mathbf{\Sigma}^{\star}$. Thus $\mathcal{E}^{\mathrm{bow}}(\mathbf{\Sigma}^{\star})$ is a finite set of bow-free SEMs that are observationally equivalent to the truth. Algebraic results on acyclic mixed graphs establish this finiteness under more general conditions (Nowzohour et al., 2017; van Ommen & Mooij, 2017).

**Minimal bow-free representations.** As multiple bow-free graphs may yield the same covariance, we seek a canonical representative. Following Deng et al. (2024), we select the sparsest. Define the edge count $s(\mathbf{B}, \boldsymbol{\Omega}) := \big|\mathrm{supp}(\mathbf{B})\big| + \big|\mathrm{supp}_{\mathrm{off}}(\boldsymbol{\Omega})\big|$, and the *minimal bow-free equivalence class*

$$\mathcal{E}^{\mathrm{bow}}_{\min}(\boldsymbol{\Sigma}^{\star}) := \big\{ (\mathbf{B}, \boldsymbol{\Omega}) \in \mathcal{E}^{\mathrm{bow}}(\boldsymbol{\Sigma}^{\star}) : s(\mathbf{B}, \boldsymbol{\Omega}) = \min_{(\mathbf{B}', \boldsymbol{\Omega}') \in \mathcal{E}^{\mathrm{bow}}(\boldsymbol{\Sigma}^{\star})} s(\mathbf{B}', \boldsymbol{\Omega}') \big\}.$$

This generalizes the sparsest Markov representation of Deng et al. (2024) from DAGs to bow-free mixed graphs.

**Assumption 2.8** (Sparsest representation). The true parameters satisfy $(\mathbf{B}^{\star}, \boldsymbol{\Omega}^{\star}) \in \mathcal{E}^{\mathrm{bow}}_{\min}(\boldsymbol{\Sigma}^{\star})$.

Under Assumptions 2.6–2.8, the target of estimation is the minimal bow-free equivalence class $\mathcal{E}^{\mathrm{bow}}_{\min}(\boldsymbol{\Sigma}^{\star})$. Within each member graph, parameters are uniquely determined by Theorem 2.5; across graphs, sparsity-based penalties select among finitely many equivalent representations.

**Penalized likelihood as a selector of minimal representatives.** Penalized likelihood provides a natural mechanism for selecting sparsest representatives within an equivalence class. At the population level, consider the negative log-likelihood $\ell(\mathbf{B}, \boldsymbol{\Omega}) := -\mathbb{E}_{\boldsymbol{\Sigma}^{\star}}\big[\log p_{\mathbf{B}, \boldsymbol{\Omega}}(X)\big]$, and a sparsity-inducing penalty $P(\mathbf{B}, \boldsymbol{\Omega})$ on both directed and bidirected edges. In this paper, we instantiate $P$ as an $\ell_1$ penalty in our implemented DECOR estimator (Section 4) and use it to derive local statistical guarantees. We conjecture that if $P$ is chosen as a quasi-MCP penalty (Deng et al., 2024) and bow-freeness is enforced as a hard constraint, then any global minimizer of the population objective $J_{\lambda, \delta}(\mathbf{B}, \boldsymbol{\Omega}) = \ell(\mathbf{B}, \boldsymbol{\Omega}) + P_{\lambda, \delta}(\mathbf{B}, \boldsymbol{\Omega})$ over bow-free SEMs belongs to $\mathcal{E}^{\mathrm{bow}}_{\min}(\boldsymbol{\Sigma}^{\star})$ for sufficiently small $(\lambda, \delta)$. Establishing this formally is beyond the scope of the present work. Such a result would extend the minimal-equivalence-class guarantee of Deng et al. (2024) from diagonal-noise DAGs to bow-free mixed graphs with latent confounding.

In the present work, we do *not* implement the full quasi-MCP estimator; instead, we focus on the $\ell_1$-penalized DECOR estimator and analyze its local convergence behavior around a fixed $(\mathbf{B}^{\star}, \boldsymbol{\Omega}^{\star}) \in \mathcal{E}^{\mathrm{bow}}_{\min}(\boldsymbol{\Sigma}^{\star})$. The equivalence-class perspective clarifies what is and is not identifiable from observational data alone, and makes explicit how bow-freeness and sparsity interact: bow-freeness and the eigenvalue margin collapse parameter ambiguity *within* each mixed graph (Theorem 2.5), while sparsity-based penalties select *among* finitely many equivalent bow-free graphs (Assumption 2.8). In Section 5, we show how these structural properties translate into curvature conditions that enable local convergence and support recovery for DECOR.

## 3 RELATED WORK

### 3.1 DAG DISCOVERY VIA CONTINUOUS OPTIMIZATION

Classical approaches to causal discovery include constraint-based methods such as PC and FCI (Spirtes et al., 2000a) and score-based procedures like GES (Chickering, 2002a). Current researches have also proposed computationally faster constraint-based causal discovery methods (Colombo et al., 2012; Bernstein et al., 2020; Shiragur et al., 2024; Pal et al., 2025). More recently, continuous optimization has emerged as a powerful alternative. NOTEARS (Zheng et al., 2018) introduced a differentiable acyclicity constraint, allowing gradient-based optimization to recover sparse DAGs. Follow-up work refined this paradigm through alternative characterizations of acyclicity (Bello et al., 2022), nonlinear extensions (Yu et al., 2019), and sparsity-regularized likelihoods such as GOLEM (Ng et al., 2020).

Despite their success, these methods generally recover graphs only up to Markov equivalence and assume causal sufficiency. Concerns have also been raised about spurious optima and reliance on data-specific artifacts (Reisach et al., 2021; Seng et al., 2023). Recent results address these issues by showing that carefully regularized scores can recover the sparsest representative of the equivalence class under mild conditions (Deng et al., 2024), but most approaches remain limited to the confounder-free case.

### 3.2 DECONFOUNDING IN CAUSAL DISCOVERY

The second line adopts a *deconfounding-first* strategy: estimate latent structure, remove its effect, then learn a DAG on residuals. Concretely, one may recover a low-rank confounding component

alongside a sparse conditional graph (Frot et al., 2019; Shah et al., 2020), fit approximate factor models under pervasiveness to extract latent scores (Wang & Blei, 2019; Squires et al., 2022), or use spectral summaries to enable downstream edge orientation, as in DeCAMFounder (Agrawal et al., 2023). These pipelines are computationally attractive and work well when a few latent factors influence many observables, yet their guarantees typically hinge on pervasiveness or nonlinearity and thus do not yield global identifiability for linear Gaussian SEMs with possibly non-pervasive confounding. Moreover, they usually target only the causal graph, treating the noise covariance as a nuisance; the confounding component is estimated and subtracted rather than modeled and identified. To distinguish our contribution, we briefly detail two recent deconfounder methods.

**Low-rank plus sparse precision decomposition.** Frot et al. (2019) assume that the observed precision matrix decomposes into a sparse component that encodes conditional relations among observables and a low-rank component induced by a small number of latent factors with pervasive loadings. Under compatibility or incoherence conditions that prevent the low-rank part from mimicking sparsity (Chandrasekaran et al., 2010), together with appropriate sample-size and tuning regimes, this split is identifiable. Intuitively, few hidden variables must influence many measured variables, while the conditional graph among observables remains genuinely sparse.

**DeCAMfounder: deconfounding via additive-noise identifiability.** Agrawal et al. (2023) target identifiability by first summarizing pervasive confounding through estimated sufficient statistics of a latent factor, then orienting edges among observables using additive-noise identifiability. Concretely, the method fits nonlinear parental mechanisms with smoothness assumptions and Gaussian disturbances conditional on the confounder summary; under these functional and distributional restrictions, the causal ordering among observed variables is identifiable from the conditional law. In purely linear-Gaussian regimes, by contrast, one typically recovers only a Markov equivalence class, so nonlinearity is essential for identification in this approach.

Our work differs in both scope and assumptions: we remain in the linear Gaussian setting, allow correlated errors induced by possibly non-pervasive confounding, and obtain global parameter identifiability under bow-free structure with a uniform eigenvalue margin on the noise covariance, leading to a single continuous optimization procedure that jointly estimates the directed structure and correlated noise.

## 4 THE DECOR ESTIMATOR

DECOR estimates $(\mathbf{B}, \mathbf{\Omega})$ by minimizing the penalized negative log-likelihood $\mathcal{J}_n$ of Equation 3, followed by a post-hoc bow-free projection. We describe the objective, the alternating optimization scheme, and the bow reconciliation procedure.

### 4.1 PENALIZED OBJECTIVE AND BICONVEX STRUCTURE

DECOR minimizes $\mathcal{J}_n(\mathbf{B}, \mathbf{\Omega}) = \mathcal{L}_n(\mathbf{B}, \mathbf{\Omega}) + \lambda_B \|\mathbf{B}\|_1 + \lambda_\Omega \operatorname{pen}_\Omega(\mathbf{\Omega}) + \rho\, h(\mathbf{B})$ s.t. $\mathbf{\Omega} \succ 0$, where $\lambda_B, \lambda_\Omega, \rho > 0$ are tuning parameters, $h(\mathbf{B})$ is the smooth acyclicity surrogate of Zheng et al. (2018), and $\operatorname{pen}_\Omega$ induces sparsity in the confounding structure. We consider two instantiations: **DECOR-COV** with $\operatorname{pen}_\Omega(\mathbf{\Omega}) = \|\mathbf{\Omega}_{\mathrm{off}}\|_1$ (sparse covariance), and **DECOR-GL** with $\operatorname{pen}_\Omega(\mathbf{\Omega}) = \|\mathbf{\Theta}_{\mathrm{off}}\|_1$ where $\mathbf{\Theta} = \mathbf{\Omega}^{-1}$ (sparse precision via graphical lasso). Ignoring $h(\mathbf{B})$, the objective is biconvex: convex in $\mathbf{B}$ for fixed $\mathbf{\Omega}$, and strongly convex in $\mathbf{\Omega}$ for fixed $\mathbf{B}$. DECOR exploits this structure via alternating minimization, using an augmented-Lagrangian treatment of $h(\mathbf{B})$ to enforce acyclicity.

### 4.2 ALTERNATING OPTIMIZATION

At iteration $t$, DECOR alternates between updating $\mathbf{B}$ (with $\mathbf{\Omega}^{(t)}$ fixed) and updating $\mathbf{\Omega}$ (with $\mathbf{B}^{(t+1)}$ fixed). Let $\mathbf{E}(\mathbf{B}) = \mathbf{X} - \mathbf{X}\mathbf{B}$ denote the residuals. We iterate the following steps until the relative change in $\mathcal{J}_n$ falls below a tolerance, yielding $(\widetilde{\mathbf{B}}, \widetilde{\mathbf{\Omega}})$.

**Step 1: DAG update.** Given $\mathbf{\Theta}^{(t)} = (\mathbf{\Omega}^{(t)})^{-1}$, we update $\mathbf{B}$ by minimizing $n^{-1} \operatorname{tr}(\mathbf{E}(\mathbf{B})^\top \mathbf{\Theta}^{(t)} \mathbf{E}(\mathbf{B})) + \lambda_B \|\mathbf{B}\|_1$ subject to $h(\mathbf{B}) = 0$. Following Zheng et al. (2018), we use

an augmented-Lagrangian formulation and perform proximal-gradient steps with soft-thresholding, updating the Lagrange multiplier after each step.

**Step 2: Noise update.** Given $\mathbf{B}^{(t+1)}$, we compute the residual covariance $\widetilde{\mathbf{S}}_{\mathbf{E}} = n^{-1}\mathbf{E}(\mathbf{B}^{(t+1)})^{\top}\mathbf{E}(\mathbf{B}^{(t+1)})$ and solve a convex subproblem for $\mathbf{\Omega}$. For DECOR-COV, we minimize $\operatorname{tr}(\mathbf{\Omega}^{-1}\widetilde{\mathbf{S}}_{\mathbf{E}}) + \log \det \mathbf{\Omega} + \lambda_{\Omega}\|\mathbf{\Omega}_{\mathrm{off}}\|_1$ via proximal gradient on the positive definite cone. For DECOR-GL, we solve the graphical lasso for $\mathbf{\Theta}^{(t+1)}$ and set $\mathbf{\Omega}^{(t+1)} = (\mathbf{\Theta}^{(t+1)})^{-1}$.

### 4.3 POST-HOC BOW RECONCILIATION

The alternating updates do not enforce bow-freeness during optimization. Instead, DECOR projects the converged estimates $(\widetilde{\mathbf{B}}, \widetilde{\mathbf{\Omega}})$ onto the bow-free constraint via a local rule applied to each pair $\{i, j\}$. We first apply hard thresholding: set $\widetilde{B}_{ij} \leftarrow 0$ if $|\widetilde{B}_{ij}| < \tau_B$, and $\widetilde{\Omega}_{ij} \leftarrow 0$ if $|\widetilde{\Omega}_{ij}| < \tau_{\Omega}$ for $i \neq j$. Then, for each pair retaining both a directed edge ($\widetilde{d}_{ij} := \max\{|\widetilde{B}_{ij}|, |\widetilde{B}_{ji}|\} > 0$) and a bidirected edge ($\widetilde{\Omega}_{ij} \neq 0$), we compare scores: if $\widetilde{d}_{ij} \geq c \cdot \widetilde{r}_{ij}$ where $\widetilde{r}_{ij} := |\widetilde{\Omega}_{ij}|/\sqrt{\widetilde{\Omega}_{ii}\widetilde{\Omega}_{jj}}$, we keep the directed channel (setting $\widetilde{\Omega}_{ij} \leftarrow 0$); otherwise we keep the bidirected channel (setting $\widetilde{B}_{ij}, \widetilde{B}_{ji} \leftarrow 0$). The resulting estimates $(\widehat{\mathbf{B}}, \widehat{\mathbf{\Omega}})$ are bow-free by construction.

## 5 OPTIMIZATION GUARANTEES

We establish convergence guarantees for DECOR and show how the post-hoc bow projection yields exact support recovery. The analysis proceeds in three steps: (i) without any structural assumptions, DECOR converges to a stationary point of the nonconvex objective; (ii) under bow-freeness and the eigenvalue margin, the same conditions ensuring identifiability (Theorem 2.5), this stationary point lies within statistical error of the truth in Frobenius norm; (iii) a post-hoc bow projection then recovers the exact support. Full assumptions are stated in Appendix B.

### 5.1 CONVERGENCE TO A STATIONARY POINT

We first characterize what DECOR guarantees *without* structural assumptions on $(\mathbf{B}^{\star}, \mathbf{\Omega}^{\star})$. DECOR is an instance of proximal alternating minimization for a Kurdyka–Łojasiewicz (KŁ) objective (Bolte et al., 2014; Attouch et al., 2013).

**Proposition 5.1** (Stationarity). *Assume $\mathcal{L}_n(\mathbf{B}, \mathbf{\Omega})$ has blockwise Lipschitz-continuous gradients on $\{\mathbf{\Omega} \succeq \eta\mathbf{I}\}$ for some $\eta > 0$, and that $P_B, P_\Omega$ are proper, lower semicontinuous penalties with well-defined proximal maps (e.g., $\ell_1$ or quasi-MCP). Then the sequence $\{(\mathbf{B}^{(t)}, \mathbf{\Omega}^{(t)})\}_{t \geq 0}$ produced by DECOR satisfies: (i) $\mathcal{J}_n(\mathbf{B}^{(t)}, \mathbf{\Omega}^{(t)})$ is nonincreasing and convergent; (ii) the iterates are bounded; and (iii) the entire sequence converges to a single first-order stationary point $(\widetilde{\mathbf{B}}, \widetilde{\mathbf{\Omega}})$ of $\mathcal{J}_n$.*

Proposition 5.1 does *not* claim global optimality—as in all nonconvex DAG learning, multiple local minima may exist—but guarantees that DECOR converges to a well-defined critical point.

### 5.2 LOCAL STATISTICAL ERROR

We now quantify how close the stationary point $(\widetilde{\mathbf{B}}, \widetilde{\mathbf{\Omega}})$ is to $(\mathbf{B}^{\star}, \mathbf{\Omega}^{\star})$. The key insight is that bow-freeness and the eigenvalue margin—the structural conditions from Theorem 2.5 that ensure identifiability, also imply favorable optimization geometry. Specifically, these conditions guarantee that the population loss satisfies blockwise restricted strong convexity (RSC) with bounded cross-block Lipschitz gradients in a neighborhood of truth; see Appendix B.2. Combined with amenable regularizers (Loh & Wainwright, 2015) tuned as $\lambda_B, \lambda_\Omega \asymp \sqrt{(\log p)/n}$, standard high-dimensional arguments yield the following.

**Theorem 5.2** (Local convergence rate). *Let $(\mathbf{B}^{\star}, \mathbf{\Omega}^{\star})$ be bow-free with $\mathbf{\Omega}^{\star} \succeq m\mathbf{I}$. Under the conditions above, there exist neighborhood radius $r > 0$, contraction factor $\rho \in (0, 1)$, and statistical error $\delta_n \asymp \sqrt{(s_B \log p)/n} + \sqrt{(s_\Omega \log p)/n}$ where $s_B = |\operatorname{supp}(\mathbf{B}^{\star})|$ and $s_\Omega = |\operatorname{supp}_{\mathrm{off}}(\mathbf{\Omega}^{\star})|$, such that the following holds with probability $1 - O(p^{-c})$. If $(\mathbf{B}^{(0)}, \mathbf{\Omega}^{(0)})$ is initialized within distance $r$ of truth, then $\|\widetilde{\mathbf{B}} - \mathbf{B}^{\star}\|_F + \|\widetilde{\mathbf{\Omega}} - \mathbf{\Omega}^{\star}\|_F = O_p(\delta_n)$.*

### 5.3 EXACT STRUCTURE RECOVERY VIA BOW PROJECTION

Finally, we formalize the post-hoc bow reconciliation (Section 4.3). For each pair $\{i, j\}$, define $d_{ij}^\star := \max\{|B_{ij}^\star|, |B_{ji}^\star|\}$ and $r_{ij}^\star := |\Omega_{ij}^\star|/\sqrt{\Omega_{ii}^\star \Omega_{jj}^\star}$. Bow-freeness implies at most one of these is nonzero for each pair. We require a margin separating the two channels.

**Assumption 5.3** (Pairwise margin). For fixed $c > 0$, define $\Delta_{ij}^\star := |d_{ij}^\star - c\,r_{ij}^\star|$ and assume $\Delta^\star := \min_{\{i,j\}} \Delta_{ij}^\star > 0$.

**Lemma 5.4** (Bow projection consistency). *Let $(\widetilde{\mathbf{B}}, \widetilde{\mathbf{\Omega}})$ satisfy $\|\widetilde{\mathbf{B}} - \mathbf{B}^\star\|_\infty \leq \varepsilon_B$ and $\|\widetilde{\mathbf{\Omega}} - \mathbf{\Omega}^\star\|_\infty \leq \varepsilon_\Omega$. Under Assumption 5.3, if $\varepsilon_B + cL\varepsilon_\Omega < \frac{1}{2}\Delta^\star$ for a constant $L > 0$ depending on the eigenvalue bounds, and post-hoc thresholds $(\tau_B, \tau_\Omega)$ are chosen appropriately, then the bow projection recovers the correct channel for every pair $\{i, j\}$, and the projected estimate $(\widehat{\mathbf{B}}, \widehat{\mathbf{\Omega}})$ exactly recovers* $\mathrm{supp}(\mathbf{B}^\star)$ *and* $\mathrm{supp}_{\mathrm{off}}(\mathbf{\Omega}^\star)$.

Combining Theorem 5.2 with Lemma 5.4 yields the main structure recovery guarantee.

**Corollary 5.5** (Exact support recovery). *Under the assumptions of Theorem 5.2 and Assumption 5.3, if the minimum signal strengths $\min_{(i,j) \in \mathrm{supp}(\mathbf{B}^\star)} |B_{ij}^\star|$ and $\min_{(i,j) \in \mathrm{supp}_{\mathrm{off}}(\mathbf{\Omega}^\star)} |\Omega_{ij}^\star|$ exceed $\sqrt{(\log p)/n}$ by a sufficient margin, then $\mathrm{supp}(\widehat{\mathbf{B}}) = \mathrm{supp}(\mathbf{B}^\star)$ and $\mathrm{supp}_{\mathrm{off}}(\widehat{\mathbf{\Omega}}) = \mathrm{supp}_{\mathrm{off}}(\mathbf{\Omega}^\star)$ with probability tending to one.*

## 6 EXPERIMENTS

We evaluate our method through comprehensive simulation studies and real-world datasets. The simulations systematically vary key structural parameters to assess identifiability and recovery performance across different regimes. As baselines, we compare against NOTEARS, GHOLE, GES, and DECAMF. DECAMF is designed to remove pervasive confounding effects and, in the linear setting, reduces to a two-step procedure: first removing a few principal components to eliminate low-rank latent structure, and then applying a structure learning method to the residualized data to estimate the sparse causal graph. For consistency and fair comparison, we employ NOTEARS in the second step.

We generate linear SEMs following the model in equation 1 with sparse directed edges $\mathbf{B}$ and low-rank-plus-diagonal noise $\mathbf{\Omega} = \mathbf{L}\mathbf{L}^\top + \sigma^2 \mathbf{I}$. The generation process ensures bow-freeness through explicit cleanup: for any $(i, j)$ pair where both $B_{ij} \neq 0$ and $\sum_k L_{ik}L_{jk} \neq 0$, we prioritize $B_{ij}$ by zeroing out the common factor loadings in row $j$. For each configuration, we sample directed edges with $B_{ij} \sim \mathrm{Uniform}([0.3, 0.8]) \times \mathrm{sign}(\mathrm{Rademacher})$ for randomly selected upper-triangular entries with density $B_{\mathrm{density}}$, generate factor loadings where each column $\mathbf{L}_{:,k}$ has $\lfloor p \cdot L_{\mathrm{density}} \rfloor$ non-zero entries drawn from $\mathcal{N}(0, 0.15^2)$, and generate data $\mathbf{X} \sim \mathcal{N}(\mathbf{0}, \mathbf{\Sigma})$ where $\mathbf{\Sigma} = (\mathbf{I} - \mathbf{B})^{-1}\mathbf{\Omega}(\mathbf{I} - \mathbf{B})^{-\top}$. We set $\sigma^2 = 0.15$ throughout to maintain a consistent eigenvalue margin per Assumption **??**. For each setting in each scenario, we generate 10 independent replicates. Unless specified otherwise, the sample complexity follows $n/p = 10$. We evaluate all methods on 10 independent replicates per density level, reporting mean performance with standard error bars.

We examine how latent confounding density affects causal structure recovery. We fix $p=20$ variables with structural density $B_{\mathrm{density}}=0.1$, assume $q=5$ latent confounders, use $n=200$ samples, and vary $L_{\mathrm{density}} \in \{0.0, 0.2, 0.4, 0.6, 0.8\}$ controlling the fraction of variable pairs influenced by shared latent factors. We compare DECOR_ADAPTIVE and DECOR_GL_ADAPTIVE (which adapt confounding regularization based on $L_{\mathrm{density}}$) against ADASCORE, NOTEARS, GOLEM, GES, LINGAM, and DECAMF_LIN variants. All methods use $\ell_1$ penalty $\lambda_B=0.1$ for structure learning; adaptive methods adjust $\lambda_\Omega/\lambda_\Theta \in \{1.0, 0.5, 0.25, 0.1, 0.05\}$ corresponding to increasing $L_{\mathrm{density}}$.

**Performance Analysis.** Figure 2 reveals several critical insights. First, *adaptive joint modeling dramatically outperforms sequential deconfounding*: DECOR_ADAPTIVE and ADASCORE achieve remarkably low SHD (20–25) and FPR ($\sim$0.01) across all densities, while DECAMF_LIN_rTrue exhibits catastrophic failure with near-zero TPR and F1 scores below 0.1. This 3–6$\times$ SHD advantage confirms that joint estimation of $\mathbf{B}$ and $\mathbf{\Omega}$ is fundamentally superior to two-stage factor-analytic residualization, which destroys causal signal.

Second, *DECOR_GL_ADAPTIVE exhibits density-dependent instability*: while achieving competitive performance at moderate densities ($L_{\mathrm{density}}=0.2$–0.4) with F1 $\approx$0.3, it suffers catastrophic

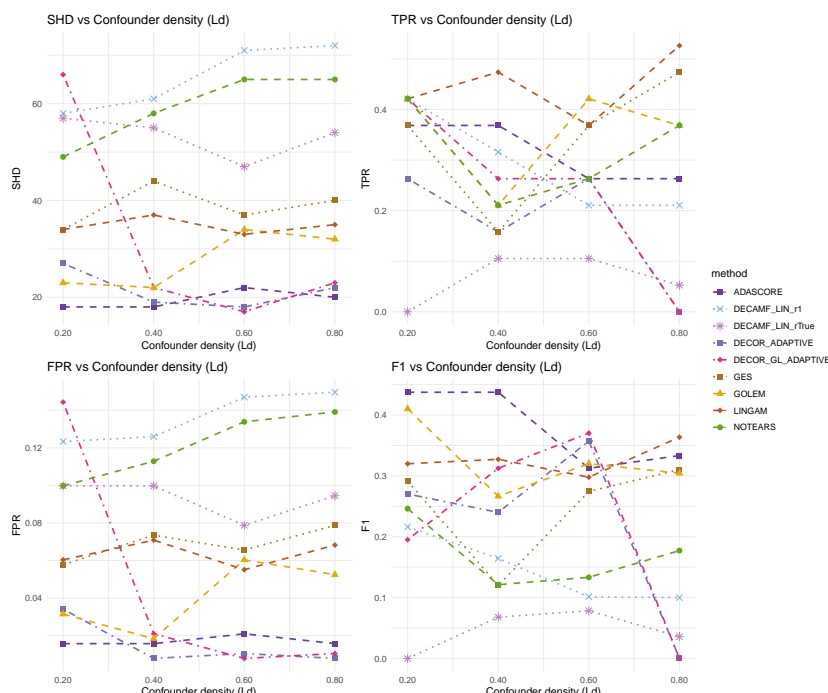

**Figure 2: Performance under varying confounding density** ($p$=20, $q$=5, $n$=200, $B_{\text{density}}$=0.1). DECOR_ADAPTIVE and ADASCORE achieve 3–6× lower SHD than traditional methods; DECOR_GL_ADAPTIVE shows instability at extreme densities. Each curve shows mean across 10 replicates; error bars indicate standard errors.

degradation at high density ($L_{\text{density}}$=0.8), where TPR drops to nearly zero and F1 collapses. The U-shaped SHD curve (starting at 125 for $L_{\text{density}}$=0.0, dropping to 20–30 at moderate densities, maintaining 30 at high density) suggests the graphical lasso approach struggles in both unconfounded and heavily confounded regimes, likely due to ill-conditioning of the precision matrix $\Theta$ when confounding structure becomes dense.

Third, *traditional methods show graceful but significant degradation*: NOTEARS, GOLEM, GES, and LINGAM maintain relatively stable SHD (50–70) and moderate TPR (0.3–0.4) as confounding increases, but their consistently higher FPR (0.1–0.15) and lower F1 (0.1–0.3) demonstrate the cost of ignoring latent confounding. These methods still recover a meaningful subset of true edges but incur substantial false discoveries, confirming theoretical predictions that unmodeled confounders induce spurious conditional dependencies.

Fourth, the *precision-recall tradeoff* distinguishes method classes: DECOR_ADAPTIVE achieves optimal balance with moderate TPR ($\sim$0.25–0.27) but exceptionally low FPR ($\sim$0.01), yielding highest F1 ($\sim$0.45). Traditional methods exhibit inverted tradeoffs—higher TPR but 10× higher FPR—reflecting liberal edge declaration when confounding creates spurious correlations. The variance across replicates (error bars) is notably lower for DECOR_ADAPTIVE than constraint-based methods (GES), reflecting continuous optimization stability.

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

Hedy Attouch, Jérôme Bolte, Patrick Redont, and Antoine Soubeyran. Proximal alternating minimization and projection methods for nonconvex problems: An approach based on the kurdyka-

