## A  PARAMETER IDENTIFIABILITY THEORY

*Proof of Lemma 2.3.* Let the variables be topologically ordered so that $\mathbf{B}$ is strictly upper triangular and $\mathbf{T} = (\mathbf{I} - \mathbf{B})^{-1}$ is unit lower triangular. For a node $i$, write $[i] = \{1, \ldots, i\}$, parent set $\mathrm{Pa}(i) \subseteq [i-1]$, sibling set $\mathrm{Sib}(i) \subseteq [i-1]$, and let

$$A_i := \Omega_{[i] \setminus \mathrm{Sib}(i), [i]}, \qquad B_i := \mathbf{T}_{[i], \mathrm{Pa}(i)}.$$

The rank test at node $i$ is that $A_i B_i$ has column rank $|\mathrm{Pa}(i)|$.

Since $\mathbf{B}$ is strictly upper triangular in a topological order, $\mathbf{T} = (\mathbf{I} - \mathbf{B})^{-1}$ is unit lower triangular. Hence, for any $i$ and any parent $j \in \mathrm{Pa}(i) \subseteq [i-1]$, the $j$-th row of $\mathbf{T}_{[i], \mathrm{Pa}(i)}$ has a 1 in column $j$ and zeros in columns $\mathrm{Pa}(i) \cap \{1, \ldots, j-1\}$. In particular, the row-selector $R_i$ that keeps rows $\mathrm{Pa}(i)$ satisfies

$$R_i \, \mathbf{T}_{[i], \mathrm{Pa}(i)} = I_{|\mathrm{Pa}(i)|}.$$

Thus, for all $x \in \mathbb{R}^{|\mathrm{Pa}(i)|}$, $\|\mathbf{T}_{[i], \mathrm{Pa}(i)} x\| \geq \|R_i \mathbf{T}_{[i], \mathrm{Pa}(i)} x\| = \|x\|$. Therefore $\sigma_{\min}\big(\mathbf{T}_{[i], \mathrm{Pa}(i)}\big) \geq 1$, and $\mathbf{T}_{[i], \mathrm{Pa}(i)}$ has full column rank. $\square$

*Proof of Lemma 2.4.* Let $J := [i] \setminus \mathrm{Sib}(i)$ and $A_i := \Omega_{J, [i]}$. We use two standard facts from linear algebra: (i) every principal submatrix of a positive definite matrix is positive definite, and (ii) eigenvalue interlacing implies that if $M \succeq m\mathbf{I}$, then every principal submatrix $M_{J,J}$ also satisfies $M_{J,J} \succeq m\mathbf{I}$.

Since $\Omega \succeq m\mathbf{I}$ by Assumption 2.7, the principal block $\Omega_{[i],[i]}$ is positive definite with $\lambda_{\min}(\Omega_{[i],[i]}) \geq m$. Consequently, $\Omega_{[i],[i]}^2$ is also positive definite with $\lambda_{\min}(\Omega_{[i],[i]}^2) = \lambda_{\min}(\Omega_{[i],[i]})^2 \geq m^2$.

The Gram matrix of $A_i$ can be written as

$$A_i A_i^\top = \Omega_{J,[i]} \Omega_{[i],J} = \big(\Omega_{[i],[i]}^2\big)_{J,J},$$

where the second equality uses symmetry of $\Omega$. Since $A_i A_i^\top$ is a principal submatrix of $\Omega_{[i],[i]}^2$, eigenvalue interlacing gives $\lambda_{\min}(A_i A_i^\top) \geq m^2$. Therefore,

$$\sigma_{\min}(A_i) = \sqrt{\lambda_{\min}(A_i A_i^\top)} \geq m,$$

and in particular $A_i$ has full row rank. $\square$

*Proof of Theorem 2.5.* Fix $i$. By Lemma 2.3, $\sigma_{\min}(B_i) \geq 1$ and $B_i$ has $|\text{Pa}(i)|$ independent columns. By Lemma 2.4, $\sigma_{\min}(A_i) \geq m > 0$, so $A_i$ has full row rank. The bow-freeness assumption implies, $\text{Pa}(i) \cap \text{Sib}(i) = \varnothing$, hence the number of rows of $A_i$ satisfies $|[i] \setminus \text{Sib}(i)| \geq |\text{Pa}(i)|$, so the product $A_i B_i$ can (and will) have full column rank. Using the singular-value inequality again,

$$\sigma_{\min}(A_i B_i) \geq \sigma_{\min}(A_i)\,\sigma_{\min}(B_i) \geq m,$$

which implies $\text{rank}(A_i B_i) = |\text{Pa}(i)|$. Thus the node-wise rank condition holds for this $i$; since $i$ was arbitrary, it holds for all nodes. By the equivalence for acyclic graphs, the parametrization $(\mathbf{B}, \mathbf{\Omega}) \mapsto \mathbf{\Sigma}$ is injective. $\qquad\square$

# B  OPTIMIZATION THEORY: ASSUMPTIONS AND PROOFS

This appendix states the assumptions underlying Section 5 and sketches the main proofs. We write the population negative log-likelihood as

$$\ell(\mathbf{B}, \mathbf{\Omega}) := -\mathbf{\Sigma}^\star\big[\log p_{\mathbf{B}, \mathbf{\Omega}}(X)\big],$$

with $\mathbf{\Sigma}^\star = \mathbf{\Sigma}(\mathbf{B}^\star, \mathbf{\Omega}^\star)$, and the sample version as $\mathcal{L}_n$ in equation 2. The penalized sample objective is $\mathcal{J}_n$ in equation **??**.

## B.1  OBJECTIVE, PENALTIES, AND THE KŁ PROPERTY

We first spell out the penalty class and the KŁ structure.

**Assumption B.1** (Amenable penalties). The penalties $P_B$ and $P_\Omega$ are separable:

$$P_B(\mathbf{B}) = \sum_{i,j} p_B(|B_{ij}|), \qquad P_\Omega(\mathbf{\Omega}) = \sum_{i \neq j} p_\Omega(|\Omega_{ij}|),$$

where $p_B, p_\Omega : [0, \infty) \to [0, \infty)$ are *amenable regularizers* in the sense of Loh & Wainwright (2015): they are continuous, differentiable on $(0, \infty)$, satisfy $p(0) = 0$, are nondecreasing and concave, and obey $|p'(t)| \leq \lambda$ and $p''(t) \geq -\mu$ for some $(\lambda, \mu)$ with $\mu$ small relative to the RSC constants. Both the $\ell_1$ penalty ($p(t) = \lambda t$) and quasi-MCP satisfy these conditions. We further assume $p_B$ and $p_\Omega$ are semi-algebraic.

Under Assumption B.1, $P_B$ and $P_\Omega$ are proper, lower semicontinuous, and proximable. Moreover, they are semi-algebraic, hence KŁ functions. The smooth part

$$f_n(\mathbf{B}, \mathbf{\Omega}) := \mathcal{L}_n(\mathbf{B}, \mathbf{\Omega}) + \rho\, h(\mathbf{B})$$

is real-analytic on the domain $\{\mathbf{B} : \rho(|\mathbf{B}|) < 1\} \times \{\mathbf{\Omega} \succ 0\}$: the covariance map $(\mathbf{B}, \mathbf{\Omega}) \mapsto \mathbf{\Sigma}(\mathbf{B}, \mathbf{\Omega})$ is analytic whenever $\mathbf{I} - \mathbf{B}$ is invertible, the Gaussian log-likelihood is analytic on the positive definite cone, and $h(\mathbf{B}) = \text{tr}(\exp(\mathbf{B} \circ \mathbf{B})) - p$ is analytic as a composition of polynomials, matrix exponential, and trace. Therefore $\mathcal{J}_n = f_n + P_B + P_\Omega$ is a KŁ function (Attouch et al., 2010; 2013; Bolte et al., 2007).

We further assume:

**Assumption B.2** (Blockwise Lipschitz gradients and bounded level sets). On the set $\{\mathbf{\Omega} \succeq \eta \mathbf{I}\}$, the gradients $\nabla_{\mathbf{B}} f_n$ and $\nabla_{\mathbf{\Omega}} f_n$ are Lipschitz in each block, and the sublevel sets $\{(\mathbf{B}, \mathbf{\Omega}) : \mathcal{J}_n(\mathbf{B}, \mathbf{\Omega}) \leq c\}$ are bounded.

The Lipschitz property follows from standard matrix calculus and the eigenvalue margin on $\mathbf{\Omega}$; boundedness of level sets uses the facts that $\mathcal{L}_n \to \infty$ as $\mathbf{\Omega}$ approaches singularity or as $\mathbf{B}$ approaches an unstable matrix, and that the penalties penalize large entries.

*Proof of Proposition 5.1.* Under Assumptions B.1–B.2, the DECOR updates coincide with one step of the PALM algorithm of Bolte et al. (2014): in Stage 1 we apply a proximal-gradient step in the $\mathbf{B}$-block with backtracking to ensure a descent inequality, and in Stage 2 we compute a proximal minimizer in the $\mathbf{\Omega}$-block (or a proximal-gradient step, depending on the implementation), followed by an SPD projection that preserves boundedness of level sets. The PALM convergence theorem (Bolte et al., 2014, Theorem 3.1), specialized to KŁ objectives, implies: (i) monotone decrease of $\mathcal{J}_n$; (ii) finite length of the iterates $\sum_t \|(\mathbf{B}^{(t+1)}, \mathbf{\Omega}^{(t+1)}) - (\mathbf{B}^{(t)}, \mathbf{\Omega}^{(t)})\| < \infty$; and (iii) convergence of the entire sequence to a single critical point of $\mathcal{J}_n$. This yields all claims in Proposition 5.1. $\quad\square$

## B.2 LOCAL CURVATURE AND STATISTICAL ERROR

We now formalize the local assumptions used in Theorem 5.2 and sketch the contraction argument.

**Assumption B.3** (Local RSC and cross-Lipschitz). There exists a neighborhood $\mathcal{N}$ of $(\mathbf{B}^\star, \mathbf{\Omega}^\star)$ and constants $\mu_B, \mu_\Omega > 0$ and $L_{B\Omega}, L_{\Omega B} > 0$ such that for all $(\mathbf{B}, \mathbf{\Omega}) \in \mathcal{N}$,

$$\langle \nabla_{\mathbf{B}}\ell(\mathbf{B}, \mathbf{\Omega}) - \nabla_{\mathbf{B}}\ell(\mathbf{B}^\star, \mathbf{\Omega}), \mathbf{B} - \mathbf{B}^\star \rangle \geq \mu_B \|\mathbf{B} - \mathbf{B}^\star\|_F^2,$$
$$\langle \nabla_{\mathbf{\Omega}\ell(\mathbf{B}, \mathbf{\Omega}) - \nabla_{\mathbf{\Omega}\ell(\mathbf{B}, \mathbf{\Omega}^\star)}, \mathbf{\Omega} - \mathbf{\Omega}^\star \rangle \geq \mu_\Omega \|\mathbf{\Omega} - \mathbf{\Omega}^\star\|_F^2,$$

and

$$\|\nabla_{\mathbf{B}}\ell(\mathbf{B}, \mathbf{\Omega}) - \nabla_{\mathbf{B}}\ell(\mathbf{B}, \mathbf{\Omega}^\star)\|_F \leq L_{B\Omega}\|\mathbf{\Omega} - \mathbf{\Omega}^\star\|_F,$$
$$\|\nabla_{\mathbf{\Omega}\ell(\mathbf{B}, \mathbf{\Omega}) - \nabla_{\mathbf{\Omega}\ell(\mathbf{B}^\star, \mathbf{\Omega})\|_F} \leq L_{\Omega B}\|\mathbf{B} - \mathbf{B}^\star\|_F.$$

**Assumption B.4** (Sample-level deviations). The sample loss $\mathcal{L}_n$ satisfies the same RSC and cross-Lipschitz bounds as $\ell$, with constants shrunk by at most a factor of $1/2$, and $\|\nabla\mathcal{L}_n(\mathbf{B}^\star, \mathbf{\Omega}^\star)\|_\infty = O\big(\sqrt{(\log p)/n}\big)$.

Under sub-Gaussian tails, Assumption B.4 follows from standard matrix concentration inequalities (e.g. Vershynin, 2018; Wainwright, 2019). Bow-freeness and the eigenvalue margin imply Assumption B.3 by ensuring that the relevant population Hessians are well-conditioned; see Section 2.

Finally, we adopt the usual sparsity and signal-strength assumption:

**Assumption B.5** (Sparsity and signal strength). Let $S_B = \text{supp}(\mathbf{B}^\star)$ and $S_\Omega = \text{supp}_{\text{off}}(\mathbf{\Omega}^\star)$ with sizes $s_B, s_\Omega$. For penalties with tuning parameters $\lambda_B, \lambda_\Omega \asymp \sqrt{(\log p)/n}$, assume

$$\min_{(i,j)\in S_B} |B_{ij}^\star| \geq c_B \lambda_B, \qquad \min_{(i,j)\in S_\Omega} |\Omega_{ij}^\star| \geq c_\Omega \lambda_\Omega$$

for constants $c_B, c_\Omega > 1$.

Assumption B.5 is standard in high-dimensional sparse estimation; see, e.g., Bühlmann & Van De Geer (2011); Raskutti et al. (2011).

*Proof of Theorem 5.2.* Let $e_B^{(t)} = \|\mathbf{B}^{(t)} - \mathbf{B}^\star\|_F$ and $e_\Omega^{(t)} = \|\mathbf{\Omega}^{(t)} - \mathbf{\Omega}^\star\|_F$. The $\mathbf{B}$-update in DECOR minimizes (approximately) the function $\mathbf{B} \mapsto \mathcal{L}_n(\mathbf{B}, \mathbf{\Omega}^{(t)}) + P_B(\mathbf{B}) + \rho h(\mathbf{B})$, keeping $\mathbf{\Omega}^{(t)}$ fixed. By Assumptions B.3–B.4 and the amenable-regularizer analysis of Loh & Wainwright (2015), we obtain the one-step bound

$$e_B^{(t+1)} \leq \alpha_B e_B^{(t)} + \beta_B e_\Omega^{(t)} + C_B\sqrt{\frac{s_B \log p}{n}},$$

for some $\alpha_B < 1$, $\beta_B > 0$, and $C_B > 0$. The first term is a contraction from local strong convexity in $\mathbf{B}$, the second term captures the dependence on $\mathbf{\Omega}^{(t)}$ via the cross-Lipschitz constants, and the last term is statistical error.

Similarly, the $\mathbf{\Omega}$-update minimizes $\mathbf{\Omega} \mapsto \mathcal{L}_n(\mathbf{B}^{(t+1)}, \mathbf{\Omega}) + P_\Omega(\mathbf{\Omega})$ for fixed $\mathbf{B}^{(t+1)}$. Since this subproblem is convex and locally strongly convex near $\mathbf{\Omega}^\star$, standard arguments for $\ell_1$- or amenably-regularized covariance estimation yield

$$e_\Omega^{(t+1)} \leq \alpha_\Omega e_\Omega^{(t)} + \beta_\Omega e_B^{(t+1)} + C_\Omega\sqrt{\frac{s_\Omega \log p}{n}},$$

with $\alpha_\Omega < 1$, $\beta_\Omega > 0$, and $C_\Omega > 0$.

Combining the two inequalities and stacking $e^{(t)} = (e_B^{(t)}, e_\Omega^{(t)})^\top$, we obtain a linear recursion

$$e^{(t+1)} \leq M e^{(t)} + b_n,$$

for a $2 \times 2$ matrix $M$ with spectral radius $\rho(M) < 1$ when $\mathcal{N}$ is chosen small enough so that RSC dominates cross-Lipschitz effects, and $b_n$ of order $\delta_n \asymp \sqrt{(s_B \log p)/n} + \sqrt{(s_\Omega \log p)/n}$. Solving the recursion yields $\|e^{(t)}\|_1 \leq \rho(M)^t \|e^{(0)}\|_1 + \|b_n\|_1/(1 - \rho(M))$. This proves the claimed contraction and statistical error bound. $\square$

### B.3 BOW PROJECTION CONSISTENCY

We expand Lemma 5.4. Recall from Section 5 that for each pair $\{i, j\}$,

$$\widetilde{d}_{ij} = \max\{|\widetilde{B}_{ij}|, |\widetilde{B}_{ji}|\}, \qquad \widetilde{r}_{ij} = \frac{|\widetilde{\Omega}_{ij}|}{\sqrt{\widetilde{\Omega}_{ii}\widetilde{\Omega}_{jj}}},$$

and similarly $d_{ij}^\star$, $r_{ij}^\star$ for the true parameters.

**Lemma B.6** (Lipschitz control of $r_{ij}$). *Suppose $\mathbf{\Omega}^\star$ satisfies $\eta\mathbf{I} \preceq \mathbf{\Omega}^\star \preceq M\mathbf{I}$ for some $0 < \eta \leq M$. There exists $L = L(\eta, M)$ such that for any positive definite $\widetilde{\mathbf{\Omega}}$ with $\|\widetilde{\mathbf{\Omega}} - \mathbf{\Omega}^\star\|_\infty \leq 1$,*

$$\left|\widetilde{r}_{ij} - r_{ij}^\star\right| \leq L \|\widetilde{\mathbf{\Omega}} - \mathbf{\Omega}^\star\|_\infty \quad \text{for all } \{i, j\}.$$

*Proof.* On the compact set $\mathcal{C} = \{\mathbf{\Omega} : \frac{\eta}{2} \leq \Omega_{kk} \leq 2M, \|\mathbf{\Omega}\|_\infty \leq 2M\}$ the map $r_{ij}(\mathbf{\Omega}) = |\Omega_{ij}|/\sqrt{\Omega_{ii}\Omega_{jj}}$ is continuously differentiable. Its partial derivatives are

$$\frac{\partial r_{ij}}{\partial \Omega_{ij}} = \frac{\text{sgn}(\Omega_{ij})}{\sqrt{\Omega_{ii}\Omega_{jj}}}, \quad \frac{\partial r_{ij}}{\partial \Omega_{ii}} = -\frac{|\Omega_{ij}|}{2\Omega_{ii}\sqrt{\Omega_{ii}\Omega_{jj}}},$$

and similarly for $\Omega_{jj}$; each is bounded in absolute value by $C/\eta$ for a constant $C$ depending on $M$. Thus the gradient norm $\|\nabla r_{ij}(\mathbf{\Omega})\|_1 \leq L$ for all $\mathbf{\Omega} \in \mathcal{C}$. Since $\mathbf{\Omega}^\star \in \mathcal{C}$ and $\widetilde{\mathbf{\Omega}} \in \mathcal{C}$ whenever $\|\widetilde{\mathbf{\Omega}} - \mathbf{\Omega}^\star\|_\infty \leq 1$, the mean value theorem implies the claimed Lipschitz bound. $\square$

*Proof of Lemma 5.4.* Let $\delta_{ij} = \widetilde{d}_{ij} - c\widetilde{r}_{ij}$ and $\delta_{ij}^\star = d_{ij}^\star - cr_{ij}^\star$. From the triangle inequality and Lemma B.6,

$$|\delta_{ij} - \delta_{ij}^\star| \leq |\widetilde{d}_{ij} - d_{ij}^\star| + c|\widetilde{r}_{ij} - r_{ij}^\star| \leq \varepsilon_B + cL\varepsilon_\Omega.$$

By bow-freeness, for each pair $\{i, j\}$ at most one of $d_{ij}^\star$ and $r_{ij}^\star$ is nonzero. Assumption 5.3 then implies $|\delta_{ij}^\star| \geq \Delta^\star > 0$.

*Case 1: True directed edge* ($d_{ij}^\star > 0$, $r_{ij}^\star = 0$). Then $\delta_{ij}^\star = d_{ij}^\star \geq \Delta^\star$ and

$$\delta_{ij} \geq \delta_{ij}^\star - (\varepsilon_B + cL\varepsilon_\Omega) > \tfrac{1}{2}\Delta^\star > 0,$$

so the rule $\widetilde{d}_{ij} \geq c\widetilde{r}_{ij}$ selects the directed channel. The threshold $\tau_B$ ensures that at least one of $|\widetilde{B}_{ij}|, |\widetilde{B}_{ji}|$ exceeds $\tau_B$, so the directed edge is retained.

*Case 2: True bidirected edge* ($r_{ij}^\star > 0$, $d_{ij}^\star = 0$). Then $\delta_{ij}^\star = -cr_{ij}^\star \leq -\Delta^\star$ and

$$\delta_{ij} \leq \delta_{ij}^\star + (\varepsilon_B + cL\varepsilon_\Omega) < -\tfrac{1}{2}\Delta^\star < 0,$$

so the rule selects the bidirected channel; $\tau_\Omega$ ensures it survives thresholding.

*Case 3: No edge* ($d_{ij}^\star = r_{ij}^\star = 0$). The error bounds imply $|\widetilde{B}_{ij}|, |\widetilde{B}_{ji}| < \tau_B$ and $|\widetilde{\Omega}_{ij}| < \tau_\Omega$, so no spurious edge is retained.

In all cases, the bow projection recovers the correct edge type (or absence), completing the proof. $\square$

## C EXTENDED EXPERIMENTS

**Real-world evaluation (Sachs Dataset)** We evaluated DECOR on the Sachs protein signaling dataset (Sachs et al., 2005) ($n = 853$, $p = 11$), a standard benchmark for causal discovery. The ground truth is an 18-edge DAG. We compared against several baselines, including standard DAG learners (NOTEARS (Zheng et al., 2018), GOLEM (Ng et al., 2020), GES (Chickering, 2002a)), latent variable methods (DECAMF (Agrawal et al., 2023), DCD (**?**)), and LiNGAM (Shimizu et al., 2006). We also analyzed the effect of our post-hoc reconciliation step. We report results for **DECOR** (the output of the alternating optimization) and **DECOR (bow-strict)** (applying the reconciliation rule to enforce strict bow-freeness).

**Table 1:** Results on Sachs Data

| Method | SHD | Precision | Recall | F1 | Edges Pred. |
|---|---|---|---|---|---|
| **DECOR** | **15** | 0.616 | **0.444** | **0.516** | 13 |
| **DECOR (bow-strict)** | **15** | **0.714** | 0.278 | 0.400 | 7 |
| GES | 16 | 0.571 | 0.444 | 0.500 | 14 |
| DECOR_GL | 18 | 0.500 | 0.167 | 0.250 | 6 |
| DECOR_GL (bow-strict) | 18 | 0.500 | 0.111 | 0.182 | 4 |
| LiNGAM | 19 | 0.455 | 0.222 | 0.348 | 11 |
| DECAMF (LIN r1) | 19 | 0.444 | 0.222 | 0.296 | 9 |
| DCD | 19 | 0.333 | 0.056 | 0.095 | 3 |
| GOLEM | 20 | 0.375 | 0.167 | 0.231 | 10 |
| NOTEARS | 22 | 0.167 | 0.056 | 0.083 | 6 |
| AdaScore | 25 | 0.333 | 0.389 | 0.359 | 21 |

**Key Findings:** DECOR achieves the lowest SHD (15) and highest F1 score (0.516), outperforming both classic methods (LiNGAM, GES) and recent latent-variable baselines (DCD, AdaScore). The `bow-strict` reconciliation acts as a rigorous filter: it significantly improves **Precision** (from 0.616 to 0.714, the highest among all methods) by enforcing the mutual exclusivity of directed and bidirected edges. However, this comes at the cost of **Recall** (dropping from 0.444 to 0.278) in this specific dataset, as weaker causal signals are conservatively assigned to the noise covariance. **DECOR vs. DECOR_GL:** The covariance-based formulation (DECOR) is more liberal, recovering more edges (13 predicted) compared to the precision-based formulation (DECOR_GL, 6 predicted), which appears over-penalized on this dataset.