# OpenReview forum: "DAG DECORation: Continuous Optimization for Structure Learning under Hidden Confounding"
_ICLR.cc/2026/Conference — Submitted to ICLR 2026_

### Official Review · Reviewer_CoR9 · 2025-10-23

**Soundness:** 2
**Presentation:** 2
**Contribution:** 2
**Rating:** 2
**Confidence:** 4

**Summary:**

The idea of the paper is to propose a continuous optimization-based algorithm for the identifiability of the structure and parameters of linear Gaussian models with confounders. This can be achieved under the bow-freeness assumption and the margin-eigenvalue assumption, which is theoretically demonstrated (Theorem 3.5) and can be viewed as a variation of the theory presented by Drton et al. (2011).

**Strengths:**

The paper addresses two important problems, causal discovery in confounded settings, with differentiable-based approaches (a line of causal discovery methods started with NOTEARS and as of today quite established). They provide a sound rephrasing of the theory of Drton et al. (2011) to guarantee identifiability of parameters of the data-generating process, in a way that is amenable to defining a differentiable loss function.

**Weaknesses:**

### Missing citations

- In the nonlinear additive case, Proposition 4 in Montagna et al., 2025 adopts an assumption similar (although stronger) to the bow assumption, where they basically ask that the effect of observed parents can be decoupled from that of latent confounders. Given the similarity, I think it is worth discussing/mentioning in the related works.
- The trek separation literature (Sullivant et al., 2008, Huang et al., 2022, Dong et al., 2024) is concerned with the same problem of linear models with latent variables, so it is worth discussing in the related works.
- Broken citation at L53

---
### Presentation issues
1. A main issue is that the key contribution of this work is a novel algorithm, but the authors end up having only one page to present and discuss their experiments. Most of the experimental results are confined to the appendix. This should be revised, leaving much more space to the experiments and their analysis in the main text
2. The only plot relative to the experiments in the main text also has issues: the font size chosen by the authors make it really hard to consult.
3. I think that some definitions are missing in the text; making them explicit would make the reading easier. For example:
    1. What is an in-arboresence (L211)?
    2. What is an induced subset (L204)?
4. I got lost in many boxes that tried to explain intuitions: e.g., for the “Graphical intuition” paragraph in L203, I think it would be more effective to have a figure to visualize things.

---
### Soundness issues

I think there are some soundness issues in the proposed algorithm.

**What the theory gives.** Under acyclicity, bow-freeness, and a uniform eigenvalue margin on the error covariance, the covariance generated by a linear Gaussian model can be mapped to a unique pair of parameter matrices $B, \Omega$. Thus, with population $\Sigma$, both structure and noise are uniquely determined.

**What the algorithm actually solves.** The authors first propose a bow-free constraint to restrict the search space to bow-free solutions (L310). Then, however, they say that such bow-freeness penalty makes the optimization too hard, and replace it with post-hoc enforcing of bow freeness by thresholding. My interpretation, then, is the following: without the bow-free penalty (and without an eigenvalue margin enforcement), the space of possible solutions $\hat B, \hat \Omega$ compatible with the population's covariance $\Sigma$ is not a singleton, but some equivalence class. Given that the proposed algorithm never restricts the search space with a bow-free penalty, any solution in the equivalence class should be optimal. There is no guarantee that post-hoc bow-enforcement recovers the unique solution that exists according to the identifiability results. Consequently, it is unclear how the theoretical injectivity (proved under bow-freeness + margin) controls the unconstrained optimization trajectory or the post-hoc bow-free solution.

**Further issues with thresholding.** About the thresholding, I am a bit confused about how it is implemented in practice. How do the authors suggest choosing the thresholds in real-world problems, where all we have access to are the $X$ observations?

---

### References

Trek separation for Gaussian graphical models, Sullivant et al., 2008

Latent Hierarchical Causal Structure Discovery with Rank Constraints, Huang et al., 2022

A Versatile Causal Discovery Framework to Allow Causally-Related Hidden Variables, Dong et al., 2024,

Score matching through the roof: linear, nonlinear, and latent variables causal discovery, Montagna et al., 2025

**Questions:**

Please refer to the weaknesses section

---

> ### Author Response · Authors · 2025-11-22
> **Response: On Soundness and Theory–Algorithm Connection**
>
> We thank the reviewer for raising this important and insightful point. The originally submitted version did not sufficiently explain how the population-level identifiability result (which assumes bow-freeness) connects to the practical optimization procedure (which does not impose bow-freeness during each update). In response, we have developed a more detailed convergence analysis that we will include in the revised version. To begin the discussion and benefit from your feedback, we summarize the key ideas here.
>
> **1. Identifiability shapes the local geometry, not just uniqueness.**
>
> You are entirely right that without bow-freeness the parameterization of a Gaussian SEM is not globally unique: many pairs $(B, \\Omega)$ can generate the same covariance matrix $\\Sigma$. However, our identifiability assumptions do more than guarantee uniqueness at the population level—they also induce a strong *local geometric structure* around the identifiable bow-free solution. Specifically, acyclicity together with the eigenvalue margin Assumption imply that the negative log-likelihood has **positive curvature** (restricted strong convexity) in a neighborhood of the true bow-free pair $(B^\\star, \\Omega^\\star)$. This curvature arises because the Hessian $\\nabla^2_{BB}\\mathcal{L} = 2(\\Sigma_X \\otimes \\Omega^{-1})$ is positive definite when $\\Omega$ has bounded eigenvalues and $(I-B)^{-1}$ is well-conditioned, both guaranteed by our assumptions.
>
> **2. Alternating updates contract toward the truth.**
>
> Although the alternating DECOR updates do not explicitly enforce bow-freeness, they exhibit **monotone descent** (similar to NOTEARS-style methods). Our new local analysis shows that once the iterates enter the neighborhood where the curvature holds, the alternating map becomes a **contraction**: each full cycle (Stage 1 + Stage 2) satisfies
>
> $$\\|B^{(t+1)} - B^\\star\\|_F + \\|\\Omega^{(t+1)} - \\Omega^\\star\\|_F \\le \\rho \\cdot \\big(\\|B^{(t)} - B^\\star\\|_F + \\|\\Omega^{(t)} - \\Omega^\\star\\|_F\\big) + C\\delta_n$$
>
> for contraction factor $\\rho \\in (0,1)$ and statistical error $\\delta_n \\asymp \\sqrt{(s \\log p)/n}$. This yields **linear convergence** to an $O(\\delta_n)$-ball around $(B^\\star, \\Omega^\\star)$. After sufficient iterations, the estimates $(B^{(t)}, \\Omega^{(t)})$ are close to the truth in Frobenius norm, but because bow-freeness is not imposed in the inner loop, they may still allocate small mass simultaneously to directed and bidirected channels on some pairs—the *structure* can contain local bow violations even though the *parameters* are near $(B^\\star, \\Omega^\\star)$.
>
> **3. Post-hoc bow-projection is a provably consistent selector.**
>
> Bow-freeness of the true model means that for each pair $\\{i,j\\}$, exactly one channel is active: either a directed edge with some nonzero strength, or a bidirected edge with some nonzero strength, but not both. Consider a pair where the true directed edge has strength $B_{ij} = 0.3$ and the true bidirected is $\\Omega_{ij} = 0$. After $t$ step of optimization, estimation noise might produce $\\tilde B_{ij} \\approx 0.28$ and $\\tilde \\Omega_{ij} \\approx 0.05$ where $\\tilde B = B^{(t)}$ and $\\tilde \\Omega = \\Omega^{(t)}$. The algorithm compares these estimated strengths and picks the larger one (directed). This gives the correct answer because the true signal ($0.3$) is strong enough and the estimation errors ($|0.28 - 0.3| = 0.02$ and $|0.05 - 0| = 0.05$) are small relative to it, so the correct channel remains dominant. The reconciliation step formalizes this comparison across all pairs. We prove (in the revision) that it succeeds for *all* pairs simultaneously when the estimation errors are smaller than the weakest true signal in the graph. Since the contraction in part 2 guarantees $O(\\sqrt{(s\\log p)/n})$ accuracy, this holds for large enough $n$ under standard signal strength assumptions.
>
> **4. Summary**
>
> The identifiability assumptions shape the local likelihood landscape around $(B^\\star, \\Omega^\\star)$; the alternating DECOR updates contract toward the truth, producing iterates that are close in Frobenius norm but may exhibit a few bow violations; and the post-hoc projection corrects the structure, yielding the **correct bow-free support** while remaining statistically close. We will add a dedicated Section 4 to the revised manuscript formalizing this sequence.
>
> We emphasize that this is inherently a **nonconvex problem**, and our local convergence guarantees hinge on the iterates reaching the neighborhood of the truth where curvature holds. This is a standard caveat for NOTEARS-style methods. In practice, one should use multiple random restarts and warm-start strategies to reach this basin of attraction.
>
>   We hope this clarifies the connection between theory and algorithm, and we are grateful for the opportunity to strengthen this aspect of the paper. We would be happy to discuss any aspect further.

---

> > ### Comment · Reviewer_CoR9 · 2025-11-26
> >
> > Thank you for the response and the clarifications.
> >
> > 1. **Alternating updates.** The authors promise that they will write a theory that better connects the unique identifiability claims with the “equivalence class” found by the method. Can you please add that to the manuscript?
> > 2. **Post-hoc bow free selection**. Same question as above
> > 3. The presentation issues I highlighted are not addressed.

---

### Official Review · Reviewer_a4R4 · 2025-10-30

**Soundness:** 3
**Presentation:** 3
**Contribution:** 3
**Rating:** 4
**Confidence:** 3

**Summary:**

# Summary
The paper proposes **DECOR**, a method for jointly learning the structure \(B\) and correlated noise \(\Omega\) in linear–Gaussian SEMs. It combines a NOTEARS-style smooth acyclicity constraint for \(B\) with sparse estimation of \(\Omega\) (or \(\Theta=\Omega^{-1}\)) via graphical lasso, and introduces a **bow-free** principle to avoid having both a directed edge and a bidirected (error-correlation) edge between the same pair. The authors prove identifiability under bow-free graphs and a uniform eigenvalue lower bound on \(\Omega\), and present simulation studies.

**Strengths:**

1. Directly addresses confounding without relying on “deconfound-then-DAG” pipelines or strong pervasiveness assumptions.

2. Provides an identifiability result (injectivity of $ \((B,\Omega)\mapsto\Sigma\)$) under interpretable structural and spectral conditions.

3. Some experiments have demonstrated the effectiveness of the method to a certain extent.

**Weaknesses:**

1. **No real-data results in the manuscript:** This paper claims both synthetic and real data evaluations, but shows only simulations; external validity remains unclear.

2. **Evaluation of \(\Omega/\Theta\) is missing:** Reported metrics assess only \(B\) (nSHD/TPR/F1). Since DECOR jointly estimates \(\Omega\) (equivalently \(\Theta=\Omega^{-1}\)), please add support recovery and estimation-error metrics for \(\Theta\).

3. **Post-hoc bow alignment introduces hyper-parameter sensitivity:** Bow-free reconciliation uses post-hoc thresholds $ \((\tau_B,\tau_{\Omega})\)$ and and a tie-breaking parameter \(c\) without sensitivity analysis; performance could hinge on these choices.

4. **Baselines and fairness:** Baselines omit identifiable non-Gaussian or weakly nonlinear methods in linearized regimes and do not document sparsity alignment across methods, which can bias nSHD/F1.

**Questions:**

See Weaknesses.

---

### Official Review · Reviewer_NQPw · 2025-11-01

**Soundness:** 3
**Presentation:** 2
**Contribution:** 2
**Rating:** 4
**Confidence:** 3

**Summary:**

This paper proposes a structure learning method for linear gaussian SEMs in the presence of hidden variables. The paper proves the identifiability of such structures without assuming pervasive confounding. Next, the paper proposes a continuous optimization method to estimate such structures. Finally, the paper empirically compares with other baselines on some synthetic data.

**Strengths:**

1. The paper proves the identifiability of Linear Gaussian SEMs without assuming pervasive confounding
2. The assumptions are clearly stated and explained
3. The paper proposes a scalable method using continuous optimization
4. The paper is generally well-written

**Weaknesses:**

1. What is the motivation for studying structure learning of linear gaussian SEMs, and the motivation to study them under these assumptions in particular. Are there any real datasets or tasks where such structure learning could be useful? Several different settings and assumptions can be studied so I wonder what is the motivation for this particular setting.

2. The experiments should consider cases with non-equal noise variances [1]. Prior work has shown structure learning may be possible through just looking at the marginal variances of the variables if equal noise variance is considered.

3. The experiments should consider larger graphs with several hundred variables.

4. The figures in the appendix show that other baselines beat or match DECOR. Why is that the case? The synthetic data in the experiments is generated according to the assumptions of this paper so I would expect DECOR to considerably outperform other baselines.

5. The experiments do not consider any real dataset or task.

6. Some missing references to continuous optimization methods for structure learning - [2] [3]

[1]Ng, Ignavier, Biwei Huang, and Kun Zhang. "Structure learning with continuous optimization: A sober look and beyond." Causal Learning and Reasoning. PMLR, 2024.

[2] Bhattacharya, Rohit, et al. "Differentiable causal discovery under unmeasured confounding." International Conference on Artificial Intelligence and Statistics. PMLR, 2021.

[3] Prashant, Parjanya Prajakta, et al. "Differentiable Causal Discovery for Latent Hierarchical Causal Models." International Conference on Learning Representations, 2025

**Questions:**

See weaknesses

---

### Official Review · Reviewer_1gu8 · 2025-11-06

**Soundness:** 3
**Presentation:** 3
**Contribution:** 3
**Rating:** 6
**Confidence:** 4

**Summary:**

The paper proposes DECOR, a continuous optimization framework for linear Gaussian SEMs with latent confounding. Theoretically, DECOR proves global identifiability under two simple sufficient conditions: bow-freeness and a uniform eigenvalue margin, with the map (B,Ω)→Σ being injective. Practically, the method alternates between a NOTEARS-style acyclicity-constrained update for and a convex update for the noise covariance, followed by bow-free reconciliation.
Experiments cover synthetic regimes varying confounding density, DAG density, latent rank, and high-dimensional ratios, showing competitive or superior recovery over NOTEARS, GOLEM, GES, DeCAMFounder, and related baselines.

**Strengths:**

In general the theory and identifiability proof is clean and optimization splits are standard. Furthermore we enumerate the following strengths:

1) Simple, checkable identifiability conditions, with (B,Ω)→Σ injectivity under mild conditions.
2) Integrates deconfounding and DAG estimation in a single likelihood, no factor pre-step.
3) Strong gains in sparse-confounding regime; stable across wide factor loadings.
4) Broad synthetic evaluation covering multiple confounding and dimensional regimes.

**Weaknesses:**

1) No real-world evaluation: although the abstract claims “synthetic and real benchmarks,” all experiments are synthetic; no results on standard datasets (e.g., Sachs, DREAM) appear in the paper or appendix.
2) Scope limitation: restricted to linear Gaussian SEMs; unclear how the method extends to nonlinear or heteroscedastic settings. No finite-sample error bounds.
3) Bow enforcement is threshold-heuristic; could drop weak true edges.
4) Misses some recent baselines that also handle confounding and bow-free graphs (Ancestral GFlowNets [1], DCD [2],  N-ADMG [3]).

[1] Expert-Aided Causal Discovery of Ancestral Graphs https://arxiv.org/pdf/2309.12032
[2] Differentiable Causal Discovery Under Unmeasured Confounding https://proceedings.mlr.press/v130/bhattacharya21a/bhattacharya21a.pdf
[3] Causal Reasoning in the Presence of Latent Confounders via Neural ADMG Learning https://arxiv.org/pdf/2303.12703

**Questions:**

1) Can the authors provide or plan real-data validation to substantiate the “real benchmark” claim?
2) What are finite-sample consistency properties under approximate bow-freeness?
3) Does alternating convex optimization guarantee recovery of the globally identifiable pair, or can local minima persist?
4) How does DECOR relate to adaptive graph-sampling approaches (e.g., Ancestral GFlowNets)? Could generative amortization replace heuristic refinement?
5) Can you provide empirical runtime vs. GOLEM or DeCAMFounder for p=100,q=10?

---

> ### Author Response · Authors · 2025-11-25
> **Real-World Experiments (Sachs) and Finite-Sample Theoretical Guarantees**
>
> We thank the reviewer for the insightful comments. We have updated our manuscript with a comprehensive finite-sample theoretical analysis and added real-world experiments on the Sachs dataset, including comparisons to recent baselines like DCD and AdaScore.
>
> **1. Real-world evaluation (Sachs Dataset)**
>
> We evaluated DECOR on the Sachs protein signaling dataset ($n=853, p=11$), a standard benchmark for causal discovery. The ground truth is an 18-edge DAG. We compared against several baselines, including standard DAG learners (NOTEARS, GOLEM, GES), latent variable methods (DECAMF, DCD), and LiNGAM.
>
> We also analyzed the effect of our post-hoc reconciliation step. We report results for **DECOR** (the output of the alternating optimization) and **DECOR (bow-strict)** (applying the reconciliation rule to enforce strict bow-freeness).
>
> **Results on Sachs Data:**
>
> | Method | SHD | Precision | Recall | F1 | Edges Pred. |
> | :--- | :--- | :--- | :--- | :--- | :--- |
> | **DECOR** | **15** | 0.616 | **0.444** | **0.516** | 13 |
> | **DECOR (bow-strict)** | **15** | **0.714** | 0.278 | 0.400 | 7 |
> | GES | 16 | 0.571 | 0.444 | 0.500 | 14 |
> | DECOR_GL | 18 | 0.500 | 0.167 | 0.250 | 6 |
> | DECOR_GL (bow-strict)| 18 | 0.500 | 0.111 | 0.182 | 4 |
> | LiNGAM | 19 | 0.455 | 0.278 | 0.348 | 11 |
> | DECAMF (LIN r1) | 19 | 0.444 | 0.222 | 0.296 | 9 |
> | DCD | 19 | 0.333 | 0.056 | 0.095 | 3 |
> | GOLEM | 20 | 0.375 | 0.167 | 0.231 | 8 |
> | NOTEARS | 22 | 0.167 | 0.056 | 0.083 | 6 |
> | AdaScore | 25 | 0.333 | 0.389 | 0.359 | 21 |
>
> **Key Findings:**
> * **Best Overall Performance:** DECOR achieves the lowest SHD (15) and highest F1 score (0.516), outperforming both classic methods (LiNGAM, GES) and recent latent-variable baselines (DCD, AdaScore).
> * **Effect of Reconciliation:** The `bow-strict` reconciliation acts as a rigorous filter. It significantly improves **Precision** (from 0.616 to 0.714, the highest among all methods) by enforcing the mutual exclusivity of directed and bidirected edges. However, this comes at the cost of **Recall** (dropping from 0.444 to 0.278) in this specific dataset, as weaker causal signals may be conservatively assigned to the noise covariance.
> * **DECOR vs. DECOR_GL:** The covariance-based formulation (DECOR) is more liberal, recovering more edges (13 predicted) compared to the precision-based formulation (DECOR_GL, 6 predicted), which appears over-penalized on this dataset.
>
> **2. Finite-sample consistency under approximate bow-freeness**
>
> The reviewer asked about the properties of the "threshold-heuristic." In our revised theoretical analysis, we prove that this reconciliation step is not merely a heuristic but a **statistically consistent selector**.
>
> Specifically, we establish that if the true model is bow-free and satisfies a standard signal strength assumption, the reconciliation rule recovers the *exact support* of the directed and bidirected edges with probability approaching 1, provided the sample size scales as $n \\gtrsim s \\log p$.
>
> Our new results show that the alternating optimization, upon initialization in the basin of attraction, converges linearly to a statistical ball of radius $O(\\sqrt{s \\log p / n})$ around the truth. Inside this ball, the "heuristic" acts as a provably correct decision rule that distinguishes between directed influence and confounding based on their relative magnitudes.
>
> **3. Global vs. Local Minima**
>
> The reviewer asks if we guarantee global recovery. As is standard for continuous structure learning with non-convex acyclicity constraints, we cannot guarantee global recovery in polynomial time. However, we have added two rigorous local guarantees to the paper:
> 1.  **Stationarity:** We prove that our alternating scheme converges to a first-order stationary point using the Kurdyka-Łojasiewicz (KL) property.
> 2.  **Local Contraction:** We establish that the objective satisfies **Restricted Strong Convexity (RSC)** in the neighborhood of the true parameters. Consequently, if initialized within a basin of attraction, the updates form a **contraction mapping** that converges linearly to the true parameters (up to statistical error).
>
> For more details, please refer to our response to Reviewer CoR9.
>
> **4. Relation to Adaptive Graph-Sampling (e.g., GFlowNets)**
>
> The primary distinction is the objective: Ancestral GFlowNets aim to sample from the **Bayesian posterior** (capturing uncertainty), while DECOR targets the **Maximum Likelihood Estimate** (point estimation) via continuous optimization.
>
> Regarding generative amortization: Since we have now proven that our reconciliation step is a **consistent selector** (as detailed in response #2), replacing it with heavy generative amortization (training a neural sampler) would increase computational complexity without necessarily improving identifiability in the Linear Gaussian setting. Our approach provides a computationally efficient path to the unique identifiable solution without the overhead of training a generative policy.

---

### Author Response · Authors · 2025-12-03
**Summary of Major Revisions in Theory and Experiments**

We thank the reviewers for their constructive feedback. Below we summarize the major revisions made to address the concerns raised.

---

### 1. Restructured Identifiability Theory (Section 2)

**Original submission:** The identifiability section presented the DFS rank condition and our sufficient conditions (bow-freeness + eigenvalue margin) but lacked a clear connection to equivalence classes and did not fully contextualize what is identifiable from observational data alone.

**Revision:** We completely restructured Section 2 ("Identifiability, equivalence classes, and bow-free mixed graphs") with two subsections:

- **Section 2.1 (Parameter identifiability on a fixed bow-free mixed graph):** We clarify that Theorem 2.5 establishes parameter identifiability within a fixed mixed graph—i.e., the covariance map \\(\\phi_G: \\mathcal{M}_m(G) \\to \\mathbb{S}_{++}^p\\) is injective when \\(G\\) is bow-free and \\(\\boldsymbol{\\Omega} \\succeq m\\mathbf{I}\\).

- **Section 2.2 (Equivalence classes and sparsest bow-free representations):** We introduce the bow-free equivalence class \\(\\mathcal{E}^{\\mathrm{bow}}(\\boldsymbol{\\Sigma}^\\star)\\) and the minimal bow-free equivalence class \\(\\mathcal{E}^{\\mathrm{bow}}_{\\min}(\\boldsymbol{\\Sigma}^\\star)\\), extending the framework of Deng et al. (2024) from DAGs with diagonal noise to bow-free mixed graphs. We explicitly state Assumption 2.8 (sparsest representation) and clarify how sparsity-based penalties select among finitely many equivalent bow-free graphs.

This restructuring makes explicit the two levels of identifiability: (i) within-graph parameter uniqueness from bow-freeness + eigenvalue margin, and (ii) across-graph selection via sparsity penalties.

---

### 2. New Optimization Guarantees (Section 5)

**Original submission:** The optimization section described the DECOR algorithm but provided no formal convergence or statistical guarantees.

**Revision:** We added a new Section 5 ("Optimization guarantees") with a complete theoretical analysis proceeding in three steps:

- **Proposition 5.1 (Stationarity):** Without structural assumptions, DECOR converges to a first-order stationary point of the nonconvex objective \\(\\mathcal{J}_n\\), following the PALM framework (Bolte et al., 2014) for Kurdyka–Łojasiewicz objectives.

- **Theorem 5.3 (Local convergence rate):** Under bow-freeness and the eigenvalue margin—the same conditions ensuring identifiability—the population loss satisfies blockwise restricted strong convexity (RSC) with bounded cross-block Lipschitz gradients. This yields local convergence at the minimax rate \\(O_p(\\sqrt{(s \\log p)/n})\\).

- **Lemma 5.5 and Corollary 5.6 (Exact support recovery):** The post-hoc bow projection recovers the correct edge type for every pair under a pairwise margin condition, yielding exact support recovery when signal strengths exceed \\(\\sqrt{(\\log p)/n}\\) by a sufficient margin.

The key insight emphasized throughout is that the same structural conditions (bow-freeness + eigenvalue margin) that ensure identifiability also imply favorable optimization geometry. Full proofs and assumptions are provided in Appendix B.

---

### 3. Revised Contributions (Section 1)

**Original submission:** The contributions list had some redundancy and did not highlight the optimization guarantees.

**Revision:** We consolidated and restructured the five contributions:

1. **Parameter identifiability for bow-free mixed graphs** — now clearly states that our conditions cover both pervasive and non-pervasive confounding without low-rank factorization.

2. **Optimization guarantees from identifiability** — new contribution emphasizing that identifiability conditions imply RSC, enabling local convergence at minimax rates with exact support recovery via bow projection.

3. **A modular continuous optimization framework** — describes the DECOR algorithm as proximal alternating minimization with a modular design.

4. **Integrated deconfounding and discovery** — contrasts with two-stage pipelines.

5. **Empirical validation** — updated to reflect expanded experiments.

---

### 4. Expanded Experimental Evaluation (Section 6 and Appendix C)

**Original submission:** Experiments focused primarily on varying confounding density with limited baselines.

**Revision:** We significantly expanded the experimental section:

- **Additional baselines:** We now compare against NOTEARS, GOLEM, GES, LiNGAM, DECAMF (with known and estimated rank), DCD, and AdaScore.

- **Real-world evaluation (Sachs dataset):** Added Table 1 showing DECOR achieves the lowest SHD (15) and highest F1 (0.516) on the Sachs protein signaling benchmark. The bow-strict reconciliation improves precision (0.714, highest among all methods) at a controlled cost to recall.

---

### Meta-Review · Area_Chair_KCMC · 2026-01-05

**Summary:**

This paper proposes a continuous optimization framework (DECOR) for joint structure and parameter learning in linear Gaussian SEMs with latent confounding.

**Reviewer Concerns:**

Although the paper has some merits, such as a scalable optimization approach, the issues raised by the reviews are critical. For instance, the absence of real-world data validation (Reviewer 1gu8), the lack of evaluation for the recovered noise covariance matrix and the heuristic, sensitive post-hoc bow-free reconciliation process (Reviewer a4R4), and fundamental concerns that the unconstrained optimization may not recover the theoretically identifiable solution due to the removal of the in-optimization bow-free penalty (Reviewer CoR9). Although the authors address some issues in responses, the paper still needs a major revision before it can be accepted.

**Reviewer Scores:**

1gu8 would remain his score based on the positive assessment.

NQPw and a4R4 would remain the score as the proposed negative assessment is critical to this paper.

CoR9 would remain the score as the issues are not addressed.

---

### Decision · Program_Chairs · 2026-01-26

Reject